# Validity and Reliability of the Singer Reflux Symptom Score (sRSS)

**DOI:** 10.3390/jpm15080348

**Published:** 2025-08-02

**Authors:** Jérôme R. Lechien

**Affiliations:** 1Department of Surgery, UMONS Research Institute for Health Sciences and Technology, University of Mons (UMons), B7000 Mons, Belgium; jerome.lechien@umons.ac.be; Tel.: +32-65373556; 2Department of Otolaryngology Head Neck Surgery, CHU Saint-Pierre, B1000 Brussels, Belgium; 3Department of Otolaryngology-Head & Neck Surgery, Foch Hospital, School of Medicine, UFR Simone Veil, Université Versailles Saint-Quentin-en-Yvelines (Paris Saclay University), 92150 Paris, France; 4Department of Otolaryngology, Elsan Hospital, 75008 Paris, France

**Keywords:** laryngeal, otolaryngology, otorhinolaryngology, voice, laryngopharyngeal, reflux, singing, singer, professional, dysphonia, hoarseness

## Abstract

**Objectives**: To investigate the reliability and validity of the Singer Reflux Symptom Score (sRSS), a new patient-reported outcome questionnaire documenting the severity of reflux symptoms in singing voice is proposed. **Methods**: Amateur and professional singers consulting the European Reflux Clinic for laryngopharyngeal reflux disease (LPRD) symptoms and findings were prospectively recruited from January 2022 to February 2023. The diagnosis was based on a Reflux Symptom Score (RSS) > 13 and Reflux Sign Assessment (RSA) > 14. A control group of asymptomatic singer subjects was recruited from the University of Mons. The sRSS was rated within a 7-day period to assess test–retest reliability. Internal consistency was measured using Cronbach’s α in patients and controls. A correlation analysis was performed between sRSS and Singing Voice Handicap Index (sVHI) to evaluate convergent validity. Responsiveness to change was evaluated through pre- to post-treatment sRSS changes. The sRSS threshold for suggesting a significant impact of LPRD on singing voice was determined by receiver operating characteristic (ROC) analysis. **Results**: Thirty-three singers with suspected LPRD (51.5% female; mean age: 51.8 ± 17.2 years) were consecutively recruited. Difficulty reaching high notes and vocal fatigue were the most prevalent LPRD-related singing complaints. The sRSS demonstrated high internal consistency (Cronbach-α = 0.832), test–retest reliability, and external validity (correlation with sVHI: r = 0.654; *p* = 0.015). Singers with suspected LPRD reported a significant higher sRSS compared to 68 controls. sRSS item and total scores significantly reduced from pre-treatment to 3 months post-treatment except for the abnormal voice breathiness item. ROC analysis revealed superior diagnostic accuracy for sRSS (AUC = 0.971) compared to sRSS-quality of life (AUC = 0.926), with an optimal cutoff at sRSS > 38.5 (sensitivity: 90.3%; specificity: 85.0%). **Conclusions**: The sRSS is a reliable and valid singer-reported outcome questionnaire for documenting singing symptoms associated with LPRD leading to personalized management of Singers. Future large-cohort studies are needed to evaluate its specificity for LPRD compared to other vocal fold disorders in singers.

## 1. Introduction

Laryngopharyngeal reflux disease (LPRD) is a disease of the upper aerodigestive tract resulting from the direct and/or indirect effects of gastroduodenal content reflux, inducing morphological and/or neurological changes in the upper aerodigestive tract [1]. The backflow of gastroduodenal content into the vocal folds may alter the vocal fold structure and biomechanical properties, leading to vocal fold function impairments in both speaking and singing voices [2,3]. LPRD has a high prevalence in laryngology, affecting approximately 50% of voice clinic patients [4]. As for many laryngopharyngeal conditions, LPRD is associated with non-specific symptoms, which supports the need to use validated patient-reported outcome questionnaire for improving patient management [1]. Professional voice users report a high LPRD incidence compared to the general population [5], which is attributed to their vocal demands, diet habits (late-night eating), and stress patterns [2,5,6]. Currently, no singer-reported outcome questionnaire specifically assessing LPRD-related symptoms in singing voice has yet been prospectively validated [2]. Indeed, currently available patient-reported outcome questionnaires documenting LPRD symptoms were not developed for singing voice, and some subtle symptoms found only during singing may not be detected/evaluated with existing questionnaires (e.g., the difficulty to reach high pitches, loss of voice intensity in singing). Developing such a patient-reported outcome questionnaire is important for detecting potential reflux consequences on singing voice, rating the LPRD impact on singing voice and measuring treatment outcomes for non-specific symptoms.

This study aims to investigate the reliability and validity of the Singer Reflux Symptom Score (sRSS), a newly developed patient-reported outcome questionnaire designed to document the severity of non-specific reflux symptoms in singing voice.

## 2. Materials and Methods

### 2.1. Development of Singer Reflux Symptom Score

The development of the sRSS started after a multidisciplinary panel discussion during the Congress of the International Federation of Otorhinolaryngological Societies (IFOS, Paris, 2016), in which international experts discussed dysphonia related to LPRD and the potential impact of gastroduodenal content reflux on speaking and singing voice. A summary of key singing symptoms was compiled at the end of the session, and the author of this paper (JRL) developed the French version of the sRSS after having developed and validated the Reflux Symptom Score (RSS) [7]. Six singers were invited to validate the main symptoms of sRSS and to rephrase some items. The French and English versions of the sRSS are available in Figure 1 and Figure 2. Note that the present validity study was conducted according to a checklist of recommendations and key characteristics to obtain valid and reliable patient-reported outcome measurements [7,8].

The sRSS consists of eight items, with the possibility to add a ninth item. Consistent with the RSS [7], the frequency and severity of each symptom are rated on a 5-point scale with each point of the frequency measure being precisely defined (Figure 1 and Figure 2). The severity score of each item is multiplied by the frequency score to obtain a symptom score ranging from 0 to 25. The sum of these symptom scores is calculated to obtain the final sRSS score, which ranges from 0 to 200 (225 in case of a ninth symptom). In addition to severity and frequency, sRSS evaluates the impact of symptoms on the singer’s quality of life (5-point Likert scale). The sRSS-quality of life score is calculated by the sum of each item score, ranging from 0 to 40. For this validity study, singers were invited to judge whether the questionnaire assessed all singing voice symptoms (yes/no).

The local ethics committee approved the study protocol (n°BE076201837630). All patients were invited to participate, and informed consent was obtained from those who enrolled in the study.

### 2.2. Subjects and Setting

Amateur and professional singers consulting the European Reflux Clinic (Brussels, Belgium) and Elsan Poitiers Polyclinic (Poitiers, France) for LPRD symptoms and findings were prospectively recruited from January 2022 to February 2023. The diagnosis was based on RSS > 13 [7] and Reflux Sign Assessment (RSA) > 14 [9], which were both validated criteria for the clinical diagnosis of LPRD in the general population. Nasovideolaryngostroboscopy was used for the RSA, considering oral, nasopharyngeal, laryngeal, and oral signs. A control group was composed of asymptomatic singers from the University of Mons. The 24 h hypopharyngeal–esophageal multichannel intraluminal impedance-pH testing (HEMII-pH) was not performed in singers because most feared the 24 h HEMII-pH probe tolerance, preferring to start an empirical therapeutic trial to achieve symptom relief whenever possible. Because LPRD diagnosis is based on the documentation of more than one pharyngeal reflux event at the 24 h HEMII-pH and the presence of LPRD symptoms (Dubai consensus) [1], asymptomatic individuals (no symptoms) did not undergo 24 h HEMII-pH either. The absence of LPRD symptoms (RSS > 13) may be consistent with the lack of LPRD diagnosis [1].

Singers and controls were excluded if they reported one of the following conditions: alcohol dependence, pregnancy, neurological or psychiatric illness, upper respiratory tract infection within the last month, current use of anti-reflux treatment (i.e., proton pump inhibitors (PPIs), histamine-H2 blockers, alginates and antacids), previous history of neck surgery or trauma, benign vocal fold lesions, malignancy, history of upper aerodigestive tract radiotherapy, untreated allergies, and inhaled corticosteroid-induced laryngitis (asthma).

The LPRD treatment included adherence to a standardized and validated anti-reflux diet [10], and the 3- to 6-month intake of post-meal alginates (Gaviscon Advance^®^, Reckitt Benckiser, Slough, UK) or antacid/magaldrate (Riopan^®^, Takeda, Zaventem, Belgium). Singers received a validated grid with diet recommendations and behavioral changes, which considered the patient’s personalized habits (late meals, alcohol intake after singing performances, etc.) [10].

### 2.3. Statistical Analyses

Statistical analyses were performed using the Statistical Package for the Social Sciences for Windows (SPSS version 29,0; IBM Corp, Armonk, NY, USA).

The reliability analysis included test–retest reliability and internal consistency. The sRSS was rated within a 7-day period to assess test–retest reliability with Spearman’s rank correlation coefficient. Internal consistency was assessed with the Cronbach-α for all sRSS items in patients and controls.

The validity analysis consisted of external and internal validity assessments. External validity was measured by correlations between sRSS, and the French version of the Singing Voice Handicap Index (sVHI) [11] with Spearman’s rank correlation coefficient. A statistical comparison between the sRSS of singers and asymptomatic individuals was carried out to evaluate internal validity (Mann–Whitney U test). Responsiveness to change was evaluated by comparing baseline and 3-month post-treatment sRSS scores (Wilcoxon signed-rank test). The sRSS threshold for supporting the presence and absence of LPRD-related singing symptoms was determined by receiver operating characteristic (ROC) analysis. A level of significance of *p* < 0.05 was used.

## 3. Results

Thirty-three singers with RSS > 13 and RSA > 14 were consecutively recruited from the European Reflux Clinic. The diagnosis was confirmed in two singers with one confounding laryngopharyngeal condition through 24 h HEMII-pH testing. The epidemiological, clinical, and singing characteristics are described in Table 1. There were 17 (51.5%) females and 16 (48.5%) males, respectively. The mean age was 51.8 ± 17.2 years. The tobacco, marijuana, and alcohol consumption details are described in Table 1. There were eight (24.2%) professional singers. Most singers were soloists (n = 27, 81.8%). The primary voice classifications in females and males were mezzo-soprano (n = 9, 27.3%) and tenor (n = 5, 15.2%). A small number of singers carried out cool-down exercise after singing (n = 5, 15.2%), while 15 (45.4%) started singing with a warmup. The primary musical/singing styles were classical, pop, pop-rock, and modern singing styles (Table 1). A control group of amateur singers was composed (n = 68; 50 (73.5%) females; age range: 18–32).

The prevalence of singing symptoms associated with LPRD are reported in Table 2. The difficulty to reach high notes and vocal fatigue/lack of vocal resistance were the two most prevalent complaints. The sRSS internal consistency analysis reported an adequate Cronbach-α (=0.832). Test–retest analysis was adequate for all sRSS items and total scores (Table 3). The intraclass correlation analysis reported adequate consistency for the test–retest reliability of the total sRSS (ICC: 0.736). The external analysis reported a significant positive correlation between sRSS and sVHI (r = 0.654; *p* = 0.015). The comparison between singers and individuals without LPRD symptoms (RSS < 13) demonstrated significantly higher item and total sRSS scores in LPRD singers compared to controls (Table 4), which indicated a high internal validity (Table 4). The pre- to post-treatment evolution of sRSS item, total, and QoL scores is reported in Table 4. Except for the abnormal voice breathiness item, all symptoms demonstrated a significant reduction after a 3-month treatment combining diet, lifestyle changes, and post-meal alginate (Table 4). The ROC curve of sRSS and sRSS-QoL is reported in Figure 3. The area under the curve values of sRSS and sRSS-QoL were 0.971 and 0.926, supporting the superiority of sRSS for determining the LPRD diagnosis. According to the Youden index, a sRSS > 38.5 reported a sensitivity of 90.3% and a specificity of 85.0%, respectively.

## 4. Discussion

The development of a singer-reported outcome questionnaire documenting the impact of LPRD on singing voice is warranted given the high prevalence of LPRD in singers, the potentially subtle singing symptoms primarily undetected and unattributed to LPRD, the need to precisely evaluate pre- to post-treatment changes in non-specific symptoms, and the significant impact of LPRD on singers’ careers [12,13,14,15,16].

The sRSS internal consistency analysis reported an adequate Cronbach-α (α = 0.832). The sRSS is the first clinical instrument developed prospectively for documenting LPRD-related singing symptoms, which limits the discussion of our results with similar instruments. However, in 2022, Nacci et al. proposed a revised Italian version of sVHI adapted to LPRD, combining Reflux Symptom Index outcomes with selected sVHI items [12,13]. In these two cross-sectional studies, authors reported a Cronbach’s alpha coefficient of 0.87 for all combined items, with item-total correlation coefficients for each item ranging from 0.461 to 0.670 [12]. In the same vein, the sRSS internal consistency is comparable to those of other singing patient-reported outcome questionnaires, such as the French (α = 0.80) [11], Greek (α = 0.960) [17], Italian (α = 0.853) [18], and Japanese (α = 0.981) [19] versions of the sVHI.

Our study suggests an adequate and moderate sRSS external validity regarding the significant association between sRSS and the French version of the sVHI (r = 0.654). Nacci et al. similarly observed a significant positive relationship between Reflux Symptom Index (not adapted to singers) and Singing Voice Handicap Index total score in a cohort of 159 singers (r = 0.474) [13]. Okui et al. reported a significant positive association between the Japanese sVHI and a visual analog scale score (r = 0.736) as external validity [19]. Note there was no external validity analysis in the other above-mentioned sVHI.

Item and total sRSS scores were significantly higher in singers with suspected LPRD compared to controls, which supports adequate internal validity. Internal validity refers to the extent to which the patient-reported outcome questionnaire accurately evaluates what it is intended to evaluate and ensures that observed changes or differences are truly due to the independent variable being manipulated, rather than other factors. The adequate sRSS internal validity data corroborate those of the Greek [17] and Japanese [19] sVHI studies, while Morsomme et al. did not find significant differences between patients and controls for the French sVHI [11].

The test–retest reliability measures the consistency of a clinical instrument over time when applied to the same individual. The sRSS test–retest reliability results demonstrated adequate values for items and total sRSS (r = 0.515–0.990), which corroborates the findings of some sVHI studies [11,17,18]. Giannopoulou et al. reported a correlation coefficient of 0.859 for the test–retest Greek sVHI [17]. In the French and Japanese sVHI studies, Morsomme et al. and Okui et al. reported test–retest validity coefficients of 0.878 and 0.93, respectively [11,19].

A limited number of studies evaluated pre- to post-treatment changes in sVHI scores. In this study, the sRSS demonstrated adequate responsiveness to change despite the small sample size. Cohen et al. evaluated the treatment responsiveness of the English sVHI in 30 singers, demonstrating a significant reduction in sVHI scores after treatment, and a significant correlation between Voice Handicap Index change and sVHI change (r = 0.71) from pre- to post-treatment [20]. A ROC curve analysis was conducted to define the threshold of sRSS associated with the highest sensitivity and specificity in detecting LPRD-related singing symptoms. Our analysis suggests a threshold of sRSS > 38.5 for the French version. Nacci et al. similarly investigated the ROC analysis of their reflux-adapted voice handicap index. The sensitivity and specificity of SVHI-12-LPR were 83.3% and 69.1% for the suspected diagnosis of LPRD with a cut-off score of 15 [12].

To the best of our knowledge, the sRSS is the first singer-reported outcome questionnaire prospectively developed for evaluating LPRD-associated singing symptoms, which is the primary strength of this study. The small sample size (singers), the lack of subgroup analysis (amateurs versus professionals, choristers versus soloists), and the absence of confirmation of LPRD diagnosis at the 24 h HEMII-pH are the main limitations. In clinical practice, it remains difficult to have full adherence from singers for 24 h HEMII due to their concerns about having a probe in their laryngopharynx for 24 h, potential tolerance issues, and their wish for rapid therapeutic intervention. Despite this reality, the lack of HEMII-pH monitoring limits the drawing of valid conclusions regarding the risk of having included patients with unknown confounding conditions associated with LPRD-like symptoms but no actual LPRD. Indeed, allergy [21,22,23], chronic rhinitis [24,25,26], chronic rhinosinusitis [27,28,29,30], tobacco-induced laryngitis [31], non-reflux chronic cough [32,33,34], or inhaled corticosteroid laryngitis [35] may lead to similar symptoms or findings compared with LPRD. The small number of singers is also an additional limitation in the drawing of valid conclusions.

Although the validation of new questionnaires typically requires comparison with asymptomatic individuals, future studies could evaluate the specificity of the sRSS in detecting LPRD in singers through comparative studies including various groups of singers with different voice disorders and multidimensional voice quality evaluations [36,37,38,39]. These studies are important to identify potential clinical differences across vocal fold disorders and to investigate the non-specificity pattern of most LPRD-related classical and singing symptoms, which are related to non-specific inflammation of vocal folds. Authors should discuss the results and how they can be interpreted from the perspective of previous studies and of the working hypotheses. The findings and their implications should be discussed in the broadest context possible. Future research directions may also be highlighted.

## 5. Conclusions

The sRSS is a reliable and valid singer-reported outcome questionnaire for screening potential singing symptoms associated with LPRD, documenting non-specific LPRD singing symptoms in symptomatic singers, and evaluating symptom changes through personalized management. An sRSS > 38.5 may be suggestive of LPRD, reporting high sensitivity and specificity, and can be used in clinical practice. Future large-cohort studies are needed to evaluate its specificity for LPRD compared to other vocal fold disorders in singers.

## Figures and Tables

**Figure 1 jpm-15-00348-f001:**
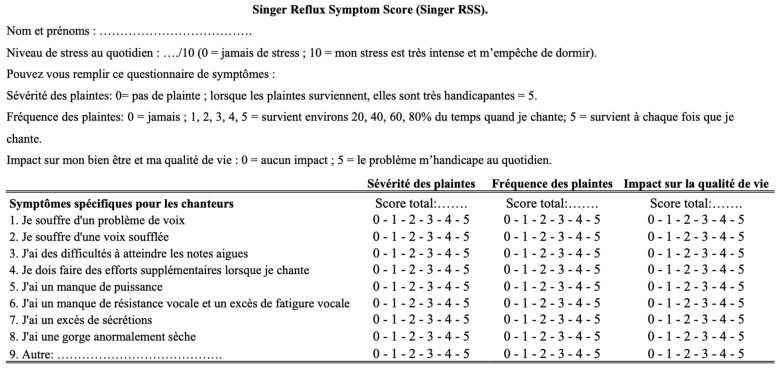
The French version of the Singer Reflux Symptom Score. The severity score of each item is multiplied by the frequency score to obtain a symptom score ranging from 0 to 25. The sum of these symptom scores is calculated to obtain the final sRSS score, which ranges from 0 to 200 (225 in case of a ninth symptom). In addition to severity and frequency, sRSS evaluates the impact of symptoms on the singer’s quality of life (5-point Likert scale). The sRSS-quality of life score is calculated by the sum of each item score, ranging from 0 to 40.

**Figure 2 jpm-15-00348-f002:**
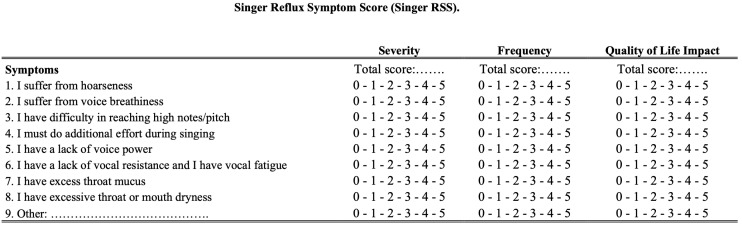
The English version of the Singer Reflux Symptom Score. The English version is not yet validated.

**Figure 3 jpm-15-00348-f003:**
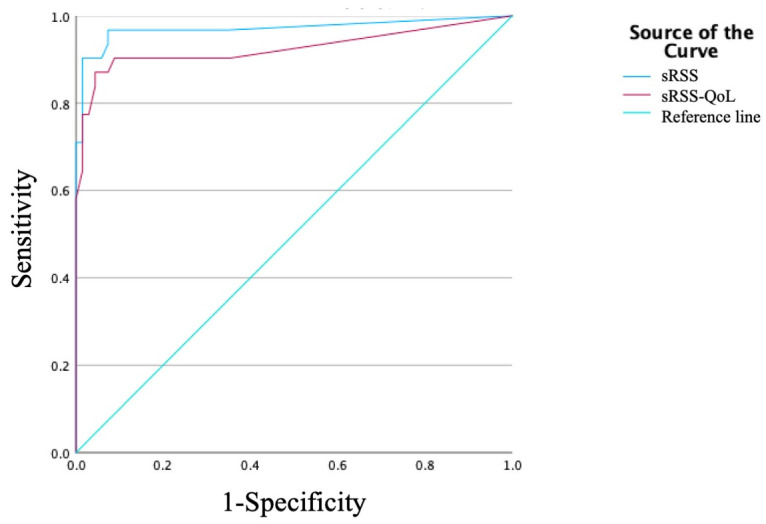
ROC Curve.

**Table 1 jpm-15-00348-t001:** Demographics, singing, and clinical outcomes of studies.

Demographics	Patients (n = 33)
Mean age (SD)	51.8 ± 17.2
Body mass index (mean, SD)	27.2 ± 8.0
Gender (N, %)	
Females	17 (51.5)
Males	16 (48.5)
Smoker (current Cig/d) and smoking history (Paq. Year)	3 (9.1)
Cannabis consumption	1 (3.0)
Alcohol (daily consumption)	3 (9.1)
Alcohol (mean and SD for unit/day in consumers)	0.2 ± 0.6
Coffee/caffeine drink/day	8 (24.2)
Singer types	
Soloist	27 (81.8)
Chorister	6 (18.2)
Voice range/classification	
Females	n = 17
Soprano	5 (15.2)
Mezzo-soprano	9 (27.3)
Alto	1 (3.0)
Several ranges	2 (6.1)
Males	n = 16
Tenor	5 (15.2)
Baritone	4 (12.1)
Bass	1 (3.0)
Several ranges	6 (18.2)
Singing habits	
Habit of performing with amplification (microphone)	2 (6.1)
Warm up before singing	15 (45.5)
Cool down after singing	5 (15.2)
Involvement in professional singing activities	8 (24.2)
Musical/Singing styles	
Classical	12 (36.4)
Pop	4 (12.1)
Rock/pop-rock	3 (9.1)
Gospel	2 (6.1)
Jazz	1 (3.0)
Opera	1 (3.0)
Modern	3 (9.1)
Several styles	7 (21.2)

Abbreviations: SD = standard deviation.

**Table 2 jpm-15-00348-t002:** Symptom prevalence in singers.

Singer Reflux Symptom Score Outcomes	Prevalence (N, %)
Hoarseness	31 (93.9)
Abnormal voice breathiness	20 (60.6)
Difficulty to reach high pitch/notes	32 (97.0)
Additional effort during singing	31 (93.9)
Lack of voice power	30 (90.9)
Vocal fatigue and lack of vocal resistance	32 (97.0)
Excess throat mucus during singing	29 (87.9)
Throat or mouth dryness during singing	25 (75.8)

**Table 3 jpm-15-00348-t003:** Test–retest reliability.

Singer Reflux Symptom Score Outcomes	Test–Retest r	*p*-Value
Hoarseness	0.647	0.001
Abnormal voice breathiness	0.990	0.001
Difficulty to reach high pitch/notes	0.630	0.001
Additional effort during singing	0.679	0.001
Lack of voice power	0.729	0.001
Vocal fatigue and lack of vocal resistance	0.763	0.001
Excess throat mucus during singing	0.516	0.001
Throat or mouth dryness during singing	0.515	0.001
Singer Reflux Symptom total Score	0.750	0.001

**Table 4 jpm-15-00348-t004:** Singer reflux symptom score internal validity and responsiveness to change.

	Baseline Evaluations				Post-Treatment (3 Months)		
Singer Reflux Symptom Score Outcomes	Patients	Controls	Z	*p*-Value	Patients	Z	*p*-Value
Hoarseness	14.9 ± 8.2	0.4 ± 3.0	−8.60	0.001	4.2 ± 5.6	−2.24	0.025
Abnormal voice breathiness	6.8 ± 8.1	0.1 ± 0.5	−6.46	0.001	1.6 ± 2.6	−1.86	0.063
Difficulty to reach high pitch/notes	15.9 ± 7.8	0.1 ± 3.3	−8.16	0.001	7.1 ± 9.2	−2.19	0.028
Additional effort during singing	13.9 ± 8.3	0.8 ± 3.3	−7.69	0.001	5.1 ± 7.1	−2.24	0.025
Lack of voice power	12.9 ± 9.2	0.7 ± 3.4	−7.99	0.001	3.7 ± 5.9	−2.52	0.012
Vocal fatigue and lack of vocal resistance	14.4 ± 8.2	0.7 ± 3.4	−8.34	0.001	5.7 ± 7.6	−2.20	0.028
Excess throat mucus during singing	11.4 ± 9.2	0.6 ± 2.7	−8.32	0.001	3.8 ± 5.5	−2.37	0.018
Throat or mouth dryness during singing	6.5 ± 7.5	0.1 ± 0.2	−6.72	0.001	1.5 ± 2.9	−2.03	0.042
Total Singer Reflux Symptom Score	96.6 ± 45.0	3.7 ± 11.0	−7.87	0.001	30.1 ± 37.4	−2.93	0.003
SRSS—Quality-of-life	21.1 ± 10.9	1.4 ± 3.1	−7.17	0.001	9.0 ± 9.4	−2.19	0.028

## Data Availability

Data are available on request.

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
