# Peer review of "Validity and Reliability of the Singer Reflux Symptom Score (sRSS)"

_jpm, 2025, doi:10.3390/jpm15080348_

Round 1
Reviewer 1 Report
Comments and Suggestions for Authors
Abstract:
- Define how the controls were recruited / selected
- Define sRSS-QoL upon first use
Introduction:
- Please, add some more evidence on why RSS is not adequate for singers. Would you use sRSS only when singing voice is affected? Or in all singers, even if their singing voice is intact, or at least not their biggest problem? This information will help understand later in the discussion, the generalizability of the results / new tool.
- To avoid confusion, please add that although “Currently, no singer-reported outcome question-naire specifically assesses LPRD-related symptoms in singing voice”, there are tools that are being developed and validated to assess the perceived impact of LPR on singing voice (such as SVHI-LPR).
Methods:
- The omission of the HEMII-pH test in all participants, including controls, is well justified, nevethereless limits the ability to confirm the presence or absence of LPRD objectively. Please, provide more information on the diagnosis, by providing evidence of the validity of the tools used and their clinical significance (e.g. were the cut-offs validated in singers, or do we get them from the general population, how did you define “asymptomatic controls”, etc).
- Some patients received alginates, abut others antacids — was this randomized, based on preference, or clinician decision? How do you expect that this affected results?
- The sRSS was rated within a 7-day period to assess test-retest reliability with the Spearman’s rank correlation coefficient: I would also expect to see an intra-class correlation analysis, which evaluates both agreement and consistency of repeated measures.
- Minor comment: I think throughout the document you mean “discriminant validity” (discrimination of between LPR and non-LPR subjects) instead of “internal validity”.
Discussion:
- It would be very useful to clarify how you intend to use this tool, e.g. as independent tool or complementary to other objective measurements. As screening tool? Diagnostic? Monitoring tool of therapy outcome? Cut-off point to know who experiences reflux and who suffers and needs treatment? In general, make it clear why we need it. It will greatly help the (clinician) reader.
- Discuss a bit more the (current) small generalizability (small sample size, narrow inclusion criteria)
Author Response
J Pers Med
Editor in chief
July, 2025.
Dear Professor
I submit the Paper entitled: “Validity and Reliability of the Singer Reflux Symptom Score (sRSS)" to JPM.
I thank the reviewer and editor for the relevant comments. All were considered.
Abstract:
-Define how the controls were recruited / selected
Done: abstract, methods, p.1, line 5: “A control group of asymptomatic singer subjects was recruited from the University of Mons.”
-Define sRSS-QoL upon first use
Done: abstract, results, p.1, last line: “ROC analysis revealed superior diagnostic accuracy for sRSS (AUC=0.971) compared to sRSS-quality of life (AUC=0.926), with optimal cutoff at sRSS>38.5 (sensitivity: 90.3%, specificity: 85.0%).”
Introduction:
-Please, add some more evidence on why RSS is not adequate for singers. Would you use sRSS only when singing voice is affected? Or in all singers, even if their singing voice is intact, or at least not their biggest problem? This information will help understand later in the discussion, the generalizability of the results / new tool.
-Requested changes done: Introduction, p.2, line 9: “Currently, no singer-reported outcome questionnaire specifically assessing LPRD-related symptoms in singing voice was prospectively validated [2]. Indeed, the current patient-reported outcome questionnaires documenting LPRD symptoms were not developed for singing voice, and some subtle symptoms found only during singing may not be detected/evaluated with the current questionnaires (e.g., the difficulty to reach high pitches, loss of voice intensity in singing). Developing such patient-reported outcome questionnaire is important for detecting potential reflux consequences on singing voice, rating the LPRD impact on singing voice and measuring treatment outcomes for non-specific symptoms.”
-Discussion: p.8, line 1: “The development of a singer-reported outcome questionnaire documenting the impact of LPRD on singing voice is warranted given the high prevalence of LPRD in singers, the potentially subtle singing symptoms primarily undetected and unattributed to LPRD, the need to precisely evaluate pre- to post-treatment changes of non-specific symptoms, and the significant impact of LPRD on singers' careers.”
-To avoid confusion, please add that although “Currently, no singer-reported outcome question-naire specifically assesses LPRD-related symptoms in singing voice”, there are tools that are being developed and validated to assess the perceived impact of LPR on singing voice (such as SVHI-LPR).
It was specified that no was -prospectively validated- to avoid confusion: introduction, p.2, line 9: “Currently, no singer-reported outcome questionnaire specifically assessing LPRD-related symptoms in singing voice was prospectively validated [2].”
Methods:
-The omission of the HEMII-pH test in all participants, including controls, is well justified, nevertheless limits the ability to confirm the presence or absence of LPRD objectively. Please, provide more information on the diagnosis, by providing evidence of the validity of the tools used and their clinical significance (e.g. were the cut-offs validated in singers, or do we get them from the general population, how did you define “asymptomatic controls”, etc).
Thank you. This information were clarified: Methods, Subjects, p.4, line 2: “The diagnosis was based on RSS>13 [7] and Reflux Sign Assessment (RSA)>14 [9], which were both validated criteria for the clinical diagnosis of LPRD in general population. A control group of asymptomatic singers was composed from the University of Mons. The 24-hour hypopharyngeal-esophageal multichannel intraluminal impedance-pH testing (HEMII-pH) was not performed in singers because most feared the 24-h HEMII-pH probe tolerance, preferring to start an empirical therapeutic trial to achieve symptom relief whenever possible. Because LPRD diagnosis is based on the documentation of more than one pharyngeal reflux event at the 24-hour HEMII-pH and the presence of LPRD symptoms (Dubai consensus) [1], asymptomatic individuals (no symptoms) did not undergo 24-hour HEMII-pH either. The absence of LPRD symptoms (RSS>13) may be consistend with the lack of LPRD diagnosis [1].”
Some patients received alginates, abut others antacids — was this randomized, based on preference, or clinician decision? How do you expect that this affected results?
Some insurances provide reimbursement of antacids. In patients with such insurance, antacids (magaldrate) were prescribed.
Antacids are chelators of bile acids/pepsin, while alginate (calcium carbonate and sodium alginate) form a raft over the stomach content and chelate as well. They are both recommended for the treatment of LPRD because act on nonacid reflux event (cf. European Consensus Guidelines for Managing LPRD). In that way, there is probably no significant differences across treatments.
The sRSS was rated within a 7-day period to assess test-retest reliability with the Spearman’s rank correlation coefficient: I would also expect to see an intra-class correlation analysis, which evaluates both agreement and consistency of repeated measures.
An additional ICC analysis was done: p.6, results, line 4: “The Intraclass correlation analysis reported adequate consistency for the test-retest reliability of the total sRSS (ICC: 0.736).”
Minor comment: I think throughout the document you mean “discriminant validity” (discrimination of between LPR and non-LPR subjects) instead of “internal validity”.
Both terms can be used according to our statistician. We used the validity properties defined in a past paper, which described all of these items: see doi: 10.1002/lary.27537.
Discussion:
-It would be very useful to clarify how you intend to use this tool, e.g. as independent tool or complementary to other objective measurements. As screening tool? Diagnostic? Monitoring tool of therapy outcome? Cut-off point to know who experiences reflux and who suffers and needs treatment? In general, make it clear why we need it. It will greatly help the (clinician) reader.
The following paragraph has been added at the conclusion for having a clinical impact: Conclusion, p.11, line 1: “The sRSS is a reliable and valid singer-reported outcome questionnaire for screening potential singing symptoms associated with LPRD, documenting non-specific LPRD singing symptoms in symptomatic singers, and evaluating symptom changes through personalized management. An sRSS>38.5 may be suggestive of LPRD, reporting high sensitivity and specificity, and can be used in clinical practice.”
-Discuss a bit more the (current) small generalizability (small sample size, narrow inclusion criteria)
Done. Discussion, p.10, line 45: “The small sample size (singers), the lack of subgroup analysis (amateurs versus professionals, choristers versus soloists), and the absence of confirmation of LPRD diagnosis at the 24-hour HEMII-pH are the main limitations. In clinical practice, it remains difficult to have full adherence from singers for 24-hour HEMII due to their concerns about having a probe in their laryngopharynx for 24 hours, potential tolerance issues, and their wish for rapid therapeutic intervention. Despite this reality, the lack of HEMII-pH monitoring limits the drawing of valid conclusions regarding the risk of having included patients with unknown confounding conditions associated with LPRD-like symptoms but no actual LPRD. The small number of singers is an additional limitation in drawing of valid conclusion, as well.”
Thanking you in advance for your attention, I remain,
Best regards,
Reviewer 2 Report
Comments and Suggestions for Authors
The presented research aim to investigate the reliability and validity of the Singer Reflux Symptom Score (sRSS), a patient-reported outcome questionnaire developed by the author.
The study design and material and methods section are well written and presented.
The main drawback of the study in my opinion is the small sample size, which is also stated in the limitations section of the paper.
It would be also beneficial to write if the diagnosis ws made only by symptom's evaluation and responsiveness to treatment or patient were undergoing nasal and endoscopic examination with fiberoscopy?
Moreover, I am a little bit concerned about the references, where 8 ot of 17 cited articles are self-citations. More variety of the existing literature should be implemented.
Author Response
#Reviewer 2:
The presented research aim to investigate the reliability and validity of the Singer Reflux Symptom Score (sRSS), a patient-reported outcome questionnaire developed by the author. The study design and material and methods section are well written and presented.
The main drawback of the study in my opinion is the small sample size, which is also stated in the limitations section of the paper.
The limitation was developed in the discussion: Discussion, p.10, line 45: “The small sample size (singers), the lack of subgroup analysis (amateurs versus professionals, choristers versus soloists), and the absence of confirmation of LPRD diagnosis at the 24-hour HEMII-pH are the main limitations. In clinical practice, it remains difficult to have full adherence from singers for 24-hour HEMII due to their concerns about having a probe in their laryngopharynx for 24 hours, potential tolerance issues, and their wish for rapid therapeutic intervention. Despite this reality, the lack of HEMII-pH monitoring limits the drawing of valid conclusions regarding the risk of having included patients with unknown confounding conditions associated with LPRD-like symptoms but no actual LPRD. The small number of singers is an additional limitation in drawing of valid conclusion, as well.”
It would be also beneficial to write if the diagnosis ws made only by symptom's evaluation and responsiveness to treatment or patient were undergoing nasal and endoscopic examination with fiberoscopy?
Details were added: Methods, p.4, line 2: “The diagnosis was based on RSS>13 [7] and Reflux Sign Assessment (RSA)>14 [9], which were both validated criteria for the clinical diagnosis of LPRD in general population. The nasovideolaryngostroboscopy was used for the RSA assessment, considering oral, nasopharyngeal, laryngeal and oral signs.”
Moreover, I am a little bit concerned about the references, where 8 ot of 17 cited articles are self-citations. More variety of the existing literature should be implemented.
The number of self-references was reduced through the removal of one non-mandatory citation and the addition of others (from other teams). However, the remaining are important in terms of methodology and discussion:
1/ The two guideline papers (Dubai – European) were led by me.
2/ The validation papers of RSS, RSA were conducted by my team, and they were used in this paper.
3/ The 3 last reviews about voice and LPRD were written by my team, as well.
4/ the standardized diet prescribed to patient was similarly mentioned as reference.
It is difficult to change these references.
Thanking you in advance for your attention, I remain,
Best regards,